# Assessment of the Maximum Amount of Orthodontic Force for PDL in Intact and Reduced Periodontium (Part I)

**DOI:** 10.3390/ijerph20031889

**Published:** 2023-01-19

**Authors:** Radu Andrei Moga, Cristian Doru Olteanu, Mircea Botez, Stefan Marius Buru

**Affiliations:** 1Department of Cariology, Endodontics and Oral Pathology, School of Dental Medicine, Iuliu Hatieganu University of Medicine and Pharmacy, Str. Motilor 33, 400001 Cluj-Napoca, Romania; 2Department of Orthodontics, School of Dental Medicine, University of Medicine and Pharmacy Iuliu Hatieganu, Cluj-Napoca, Str. Avram Iancu 31, 400083 Cluj-Napoca, Romania; 3Department of Structural Mechanics, School of Civil Engineering, Technical University of Cluj-Napoca, Str. Memorandumului 28, 400114 Cluj-Napoca, Romania

**Keywords:** maximum orthodontic force for PDL, periodontal breakdown, maximum hydrostatic pressure, ischemic and necrotic risks, orthodontic movements, finite elements analysis

## Abstract

This study examines 0.6 N and 1.2 N as the maximum orthodontic force for periodontal ligament (PDL) at multiple levels of periodontal breakdown, and the relationships with the ischemic, necrotic, and resorptive risks. Additionally, this study evaluates if Tresca failure criteria is more adequate for the PDL study. Eighty-one 3D models (from nine patients; nine models/patients) with the 2nd lower premolar and different degrees of bone loss (0–8 mm) where subjected to intrusion, extrusion, rotation, translation, and tipping movements. Tresca shear stress was assessed individually for each movement and bone loss level. Rotation and translation produced the highest PDL stresses, while intrusion and extrusion determined the lowest. Apical and middle third PDL stresses were lower than the cervical stress. In intact periodontium, the amount of shear stress produced by the two investigated forces was lower than the 16 KPa of the maximum physiological hydrostatic pressure (MHP). In reduced periodontium (1–8 mm tissue loss), the apical amount of PDL shear stress was lower than MHP for both applied forces, while cervically for rotation, translation and tipping movements exceeded 16 KPa. Additionally, 1.2 N could be used in intact periodontium (i.e., without risks) and for the reduced periodontium only in the apical and middle third of PDL up to 8 mm of bone loss. However, for avoiding any resorptive risks, in the cervical third of PDL, the rotation, translation, and tipping movements require less than 0.2–0.4 N of force after 4 mm of loss. Tresca seems to be more adequate for the study of PDL than other criteria.

## 1. Introduction

The key point in the study of the maximum amount of orthodontic force applied in intact and reduced periodontium, without any ischemic, necrotic, and resorptive risks, is the periodontal ligament (PDL), which plays the leading part in the orthodontic movement.

The periodontal ligament, a fibrous connective tissue of 70% water, has approximately 0.2 mm (ranging from 0.15 mm to 0.38 mm), a soft and dense consistence, and is involved in bone and cementum remodeling metabolism. The collagen fibers are displayed as variously orientated dense fiber bundles filing a space of 0.4–1.5 mm by connecting the bone to the cementum and dissipating the stress of various direction movements. The vascular support is well represented: apical vessels derived from pulpal supply, perforating vessels, and gingival vessels. The blood vessels facing outwards are involved in the biomechanical suspension and dissipation of pressures, while those facing inwards are involved in the nutritional metabolism of PDL tissues. Thus, due to the rich blood supply and remodeling processes, the tissue’s adaptability to various amounts of continuous and discontinuous forces during orthodontic treatment is possible, and if the optimal force is used, no ischemic, necrotic, and root resorption risks are encountered. In the PDL vascular support, the physiological hydrostatic pressure was reported to be between 2 and 16 KPa (approximately 80% of human systolic blood pressure) [1,2,3,4,5,6].

The risk of further periodontal loss and orthodontic external root resorption (both apically and cervically) in patients suffering from chronic periodontitis is always present during the orthodontic treatment [1,4,5,6,7,8]. It depends on the amount of applied orthodontic force (i.e., continuous vs. discontinuous forces, and time to maintain the force), if the maximum physiological hydrostatic pressure (MHP) of 2–16 KPa is exceeded or not (i.e., how much time the pressure in excess is kept), the variable level of periodontal breakdown present around the teeth (i.e., higher bone loss means lower capacity of dissipation and protection), and the adaptability of tissues to withstand damages [1,4,5,6,7,8,9]. There are reports about continuous orthodontic forces up to 1.2 N that could generate in intact periodontium stresses exceeding MHP, inducing ischemia and regressive changes in the structures of both teeth (e.g., dental pulp and cementum) and periodontium’s (e.g., neuro-vascular bundle—NVB, periodontal ligament—PDL, trabecular, and cortical bone) [4,5,6,7,8,9,10,11,12,13]. Orthodontic movements follow a circulatory disruption and a remodeling of the PDL under the effects of a balanced orthodontic force [1,6,7,14]. Nonetheless, the orthodontic external cervical root resorption was also reported to derive from a disruption of the periodontal ligament of pathological ischemia and with lacking any clinical signs [10].

In the scientific literature there is still no agreement regarding the quantitative value of the optimal/maximum amount of force (i.e., generally considered to be light [1,14]) and which movement is more likely to cause ischemic problems, and in which areas (e.g., the discrepancy between the values provided by different studies: some reported [2,3,15,16] up to 3–6 N for intact periodontium, with external root resorption [4,5,10,17] for tipping, translation, and intrusion, while others [1,6,7,8] reported maximum 1.2 N for no bone loss, 0.6 N for reduced periodontium, and no risk of external root resorption). Moreover, these reports [2,3,4,5,10,15,16,17] used different methods of studying, thus creating even more confusion. Nevertheless, for minimizing risks of further tissue loss, recent studies suggested that MHP should not be exceeded for a prolonged period [1,6,7,8,12,13,16] (i.e., due to adaptability of periodontal tissues to sustain damage for only a short period of time). A report of our group found the rotation movement to be the most invasive along with translation, and that 0.5 N is safe to be used for the cervical third of PDL in intact periodontium, while the amount of force should be reduced to 0.1–0.2 N for rotation, 0.15–0.3 N for translation, and 0.2–0.4 N for tipping in 4–8 mm of periodontal breakdown [1]. However, in the apical third of PDL, the same study [1] reported 0.5 N to be safely applied to up to 8 mm of periodontal loss. Nonetheless, the questions about the maximum orthodontic force to be safely used in the apical third of PDL at various levels of bone height without any ischemic and/or necrotic risks remain.

Recent review reports investigating PDL in in vivo studies considered that most clinical studies related to this subject suffer from quality problems [18,19], and are associated with other reports [1,6] of insufficient data regarding the orthodontics in reduced periodontium, which enhanced the need of more data.

Periodontal breakdown could be completely analyzed only in vitro by means of finite elements analysis (FEA) [1,6,7,8,20], allowing for the performance of individual analysis of each tooth and periodontal component under various conditions (i.e., variations in the amount of force, appliance point, variable physical properties, and different bone heights). Only a FEA study allows for following the relationships between the biomechanical behavior of the different anatomical components and performing associations with the already known clinical data [1,6,7,8,20], which is mandatory for avoiding continued tissue loss. It is accepted that the remodeling of the periodontal tissues (especially in the PDL cervical third, frequently the most affected) is caused by various levels of circulatory disturbances triggered by stresses produced by the orthodontic force and dependent on the associations among optimal force, type of movement, affected area, and level of bone loss [4,6,7,10]. Despite sustainability due to periodontal tissue’s adaptability and being partially reversible and recoverable after up to three months, the periodontal circulation should not be subjected to higher pressures than MHP for prolonged time [11,12,13]. Thus, the importance of having data from each tooth and periodontal part about the stress dissipation, the affected areas, and the highest tolerable quantitative force in different orthodontic movements is underlined. Nevertheless, little information about these issues is available when studying the periodontal breakdown.

Despite being a relatively easy and exact method of in vitro study of anatomical complex tissues, FEA is viewed with caution due to many reports with debatable quantitative results. This method is based on CBCT (cone-beam computed tomography) investigations of patients with various levels of periodontal breakdown and is widely used in clinical practice. The acquired data pass than through a reconstruction process (i.e., manual, or automated) using a special software for recognizing and finding the different levels of grey form the DICOM slices [1,6,7,8]. The result is a 3D anatomical exact model of the clinical situations that can be subjected to multiple different types of simulations [1,6,7,8].

The simulations of the biomechanical behavior of an anatomical tissue can be achieved using different FEA software. They are widely used in the engineering field (e.g., from aerospace industry to buildings and car constructions) with exact results and were introduced in the medical and dental fields in recent years. Despite analyzing the entire structure, FEA allows individual analysis of each part of the structure, thus offering unlimited studying possibilities. However, despite being considered the right method in the engineering field, in the dental field, it is regarded with mistrust due to different issues related to reports that contradict in vivo clinical data [1,4,5,6,7,8,21,22,23,24,25,26,27]. Nonetheless, if three main prerequisites are respected and satisfied, the FEA is as accurate and reliable as in the engineering filed. These prerequisites are: the employment of a material type-based criteria (for ductile or brittle materials [1,6,7,8,20]), the correct physical properties (boundary conditions), and an accurate 3D model [6,7,8,20]. In the dental studies [4,5,24,25,26,27,28,29,30,31] from the past decade, these prerequisites were neither acknowledged nor respected/satisfied. Thus, for the PDL studies (i.e., periodontal ligament considered to have a more ductile resemblance), the employed failure criteria: maximum principal S1 and minimum principal S3 (design only for brittle materials) [24,25,26,27,30,32] and hydrostatic pressure (designed only for liquids and gases) [4,5,17] were intensively used instead of the more adequate von Mises (VM) and Tresca specially designed for ductile materials [1,6,7,8,24,25,32,33,34]. As a result, their quantitative reports [4,5,24,25,26,27,28,29,30,31] exceeded MHP (considered indirect validation criteria of results) and did not perform associations with clinical data. However, recent FEA studies [1,6,7,8], by acknowledging the three main prerequisites (considered also limitations), were able to report quantitative results correlated with clinical data and performed the necessary relationships, proving the viability of the method in the study of periodontium and the feasibility in gathering information that otherwise cannot be obtained from clinical studies.

The mostly used failure criteria adequate for ductile materials (e.g., steel or rubber) are von Mises and Tresca. In the engineering field, Tresca is more restrictive and, under certain circumstances, more right than von Mises [1,6]. Our here analysis, by employing a potential more exact criterion, aims to supply evidence to clarify the problem of the best (i.e., maximum) amount of orthodontic force applied in PDL during the periodontal breakdown process.

This study examines if 0.6 N and 1.2 N could be the maximum amount of orthodontic force for PDL at multiple levels of periodontal breakdown, and the relationships with the ischemic and resorptive risks, further tissue loss, and the type of movement. As a second aim, this study examines if Tresca FEA failure criteria is more adequate for the in vitro study of the periodontal ligament.

## 2. Materials and Methods

The herein analysis is part of a bigger project (clinical protocol 158/02.04.2018) developed progressively in phases with the clear goal to study the lower premolar and the surrounding periodontium at different bone heights under orthodontic forces [1,6,7,8]. This study is focused on finding the maximum amount of orthodontic force to be applied for the apical and cervical third of PDL without any major ischemic/necrotic risks.

This analysis was performed over eighty-one 2nd lower premolar 3D models from nine patients (4 males and 5 females, mean age 29.81 ± 1.45 years, and informed oral consent) with reduced noninflamed periodontium (i.e., treated chronic periodontitis/stage II/III grade B periodontitis enrolled in supportive periodontal therapy). Even though more patients were considered for this research, only nine met the inclusion criteria: intact and reduced noninflamed periodontium, proper oral hygiene, complete mandibular arches (no teeth loss for the studied arch), intact second lower premolar (no decays, fillings, and endodontic or prosthetic treatment) with adequate anatomical topography (no malposition), various levels of bone loss, indication for orthodontic treatment and availability of follow-up throughout treatment. The exclusion criteria included: intact and reduced inflamed periodontium, poor oral hygiene, tooth loss (lower first molar and premolars), second lower premolar with decays, fillings, prosthetic and endodontic treatment, malposition, and inconsistency in the follow-up will. Taking into account that most of the patients who present themselves for orthodontic treatment do not meet the inclusion criteria (due to associated dental troubles) and having in mind that previous studies related to our subject investigated one or two models, from one patient, we considered that by using 81 3D models from nine patients, valid results and conclusions could be drawn from our study. The area of focus was analyzed using X-ray (CBCT/cone-beam computed tomography-ProMax 3DS-Planmeca, Finland; voxel size 0.075 mm) and included the lower premolars and first and second molars.

Using AMIRA 5.4.0 software (AMIRA, version 5.4.0, Visage Imaging Inc. 300 Brickstone Square, Suite 201 Andover, MA 01810, USA) for manual image segmentation, each anatomical component present in CBCT data were recreated and then reassembled into a 3D mesh model (obtaining nine mesh models with different rates of periodontal breakdown—Figure 1). For each of these nine mesh models, the missing PDL and bone were reconstructed as close as possible to the anatomical reality (5,058,673–6,047,378 C3D4 tetrahedral elements of 0.08–0.116 mm, 950,897–1,062,438 nodes). The 2nd premolar was guarded, while the rest of the teeth were replaced by bone (cortical and trabecular). Thus, nine intact periodontium mesh models were obtained and then subjected to a horizontal periodontal breakdown of 1 mm up to 8 mm of loss, obtaining 72 mesh models (Figure 1). A total of eighty-one 3D models (i.e., nine models for each patient) were analyzed in this study. Each PDL was shaped with a variable thickness of 0.15–0.225 mm and was reconstructed with its own neuro-vascular bundle (NVB), Figure 1.

ABAQUS 6.13-1 software (Dassault Systèmes Simulia Corp., Stationsplein 8-K, 6221 BT Maastricht, The Netherlands) was employed for the FEA simulation, and 0.6 N (approx. 60 g) and 1.2 N (approx. 120 g) of intrusion, extrusion, rotation, tipping, and translation were applied at the bracket level of each tooth (Figure 1). All models were subjected to similar boundary conditions, material properties, and loading conditions (Table 1). Tresca shear stresses were found and shown for the entire PDL (Table 2) and numerically expressed as a color-coded projection (Figure 2). The quantitative values were than correlated with MHP, risks of external orthodontic root resorption, ischemia/necrosis, and further periodontal loss. If the displayed amount of quantitative stress under one or both investigated orthodontic forces reach or exceed the reported maximum physiological hydrostatic pressure, induce a high risk of ischemia, necrosis, and resorption, then no higher force should be used, being the maximum amount to be tolerated by the tissue. Based on this risk assessment, the simulations with Tresca failure criterion were redone for all models by reducing the amount of orthodontic force applied to the bracket to a level of 0.1–0.4 N, and then the average quantitative results were correlated once more with MHP. Stress increase speed correlated with quantitative stress values for intact periodontium as the reference point. Quantitative results (Table 2) were also associated with those reported by two earlier simulations [1,6] of our group employing the von Mises and Tresca failure criterions for PDL.

The homogeneity, isotropy, linear elasticity, and perfectly bonded interfaces were assumed (Table 1), while all 3D models had a fixed model base.

## 3. Results

In terms of quantity, the cervical third stress was higher compared to the apical and middle third stress. The most significant quantitative stresses were displayed by the rotation and translation movements, with lower values for tipping, extrusion, and intrusion (Table 2, Figure 2). From the qualitative point of view, the extent of the stressed areas remained unchanged in both simulations (i.e., with 0.6 and 1.2 N), while only the amount changed (Figure 2 and Figure 3). The qualitative stress distribution (i.e., color-coded projections) in tissues maintained a similar resemblance for all models, while the amounts of stress (i.e., quantitative values) showed small differences from one model to another. Neither the age and gender, nor the periodontal status had any visible influence over the results.

In undamaged/intact periodontium 0.6 N of orthodontic force produced stresses lower than the 16 KPa of MHP in the entire PDL (Table 2, Figure 2), while for 1.2 N of applied force (Figure 3), only the cervical third of the PDL (i.e., only for translation, rotation, and tipping) displayed areas of stress exceeding the physiological circulatory pressure (i.e., 1.5–2.3 times) (Table 3).

In damaged/reduced periodontium, the quantitative stress produced by a continuous orthodontic force of 0.6 N (Table 2) was lower than the MHP in the apical and middle third of the PDL at up to 8 mm of bone loss for all five movements. However, in the cervical third of the PDL, a higher amount of stress appeared at 7–8 mm of the periodontal breakdown for intrusion and extrusion, while for the other movements the overcoming of tolerable physiological stress appeared after 1–2 mm of bone loss continuing up to 8 mm of tissue resorption (e.g., translation 1.4–4.2 times, rotation 1.6–2.5 times, and tipping 1.2–2.8 times). A doubling of stress at 3–4 mm of bone loss and a tripling after 7–8 mm was observed.

A continuous force of 1.2 N displayed in the apical third of reduced periodontium amounts of stress lower than the MHP for all five movements, seeming to be safe to be applied. In the middle third, the same amount of force was less than the 16 KPa for intrusion and extrusion up to 8 mm of resorption, while for the other movements remained lower, only up to 4–5 mm of loss (except for rotation, 3 mm of tissue loss). Nevertheless, for the rotational movement, a doubling of the stress was present at 8 mm of bone loss in the middle third of the PDL. The translational and tipping movements (i.e., from 4–5 to 8 mm of bone loss) displayed stresses less than double the value of the physiological pressure. The cervical third of the PDL displayed stresses lower than MHP for intrusion and extrusion up to 1–2 mm of resorption. The other three movements displayed cervical quantitative stresses exceeding the 16 KPa (the double or triple of MHP) for the first mm of height of tissue loss. The highest stress values were cervically displayed by translation and rotation (8.5 times higher) at 8 mm of periodontal breakdown.

Qualitatively, the intrusion and extrusion movements displayed stresses in the entire PDL, while translation, rotation, and tipping were displayed mainly in the cervical third (Figure 2 and Figure 3). In each of the five movements, and for the entire horizontal periodontal breakdown simulation, the highest amount of stress was displayed in the upper part of the cervical third of PDL (color coded in red and orange—Figure 2 and Figure 3), with a height of approximately 1 mm. The similar pattern was kept for both 0.6 and 1.2 N of applied orthodontic forces.

The intrusion and extrusion in intact periodontium seems to produce the highest amount of apical third stress up to 1.2 N of applied force, followed by rotation and translation, but without ischemic or resorptive risks. In the middle and cervical third of PDL, the rotation and translation movements seem to be the most invasive, while intrusion and extrusion are the least ones.

In reduced periodontium (1–8 mm of loss), the rotational movement seems to be the most invasive of all five (i.e., the highest amount of stress), for the PDL (Table 2 and Table 3), while intrusion and extrusion remain the least of them.

Based on the average quantitative stresses displayed in Table 2, 0.6 N of continuous force seems to be periodontally acceptable and tolerable (in the cervical third of PDL- due to the limited areas of stress displayed of approximately 1 mm of height, Figure 2) for up to 8 mm of periodontal breakdown. However, due to stress display (Figure 3), 1.2 N of continuous force seems to be acceptable and tolerable only in the apical and middle third of PDL, while in the cervical third, especially in rotational and translational movement (i.e., by highly exceeding the MHP values), it seems to have high ischemic, necrotic and resorptive risks. Thus, for 4–8 mm of tissue loss, the cervical third stress is less than the MHP if the applied orthodontic forces stay under 0.6 N, respectively, between 0.15 and 0.4 N (0.2–0.4 N for tipping, 0.15–0.2 N for rotation, and 0.15–0.25 N for translation).

A directly proportional correlation between bone loss and force reduction was seen up to a doubling of stress for each of the two force simulations. The quantitative results are within the acknowledged limit (approximately 15–30% higher) when comparing with VM failure criteria.

## 4. Discussion

The herein analysis assessed if 0.6 N and 1.2 N of continuous orthodontic force could be seen as the maximum amount of orthodontic force to be safely used for up to 8 mm of horizontal periodontal breakdown. For each of the two forces, associations between the maximum physiological hydrostatic pressure with the ischemic, necrotic and resorptive risks were also investigated. An examination of Tresca as a possible more adequate failure criterion to be used for the study of PDL was conducted. Our team did not find any other studies with similar aims and methodology except our earlier reports [1,6].

The two selected amounts of force of 0.6 N and 1.2 N are usually used in the clinical orthodontic therapy and close to the limits of applied forces. In a previous study [1] a simulation with 0.5 N, showed an amount of stress lower than the physiological maximum hydrostatic pressure (MHP). Thus, a higher force of 0.6 N was selected to be studied. Furthermore, FEA being a mathematical algorithm-based method, it allows a mathematical anticipation/calculation and prediction of PDL behavior to be subjected to a higher force, thus allowing the selection of 1.2 N. These two forces being commonly used by the practitioner in daily practice was considered to be of high interest. If the amount of stress produced by one or both of them exceeds the MHP, it implies that no higher force should be used, being the maximum amount to be tolerated by the tissue.

Based on the quantitative values (Table 2 and Table 3) and stress display areas (Figure 2 and Figure 3), 0.6 N of force could be considered to be relatively safe in both intact and 1 to 8 mm of reduced periodontium, with almost no risks of circulatory disturbances, confirming our previous reports [1,6], in agreement with some studies [24,25,32,33] and contradicting other reports [2,3,4,5,16]. A higher force of 1.2 N, despite being completely safe for the apical vessels of blood supply from the apical third of PDL (holding the reconstructed NVB of the tooth and dental pulp), could generate some minor ischemic risks for the perforating vessels in the middle third after 4–5 mm of bone loss, in disagreement with Hohman et al. and Wu et al. reports [2,3,4,5,16] (which employed a non-adequate failure criteria), but in agreement with previous analysis [1,6]. However, for the gingival vessels from the cervical third, the same 120 g of force seem to be generating ischemic risks, especially in 1–8 mm reduced periodontium (Figure 3). Nonetheless, in reduced periodontium for rotation, translation, and tipping, the most affected area is represented by the first 1 mm of height of the cervical third of PDL (i.e., limited areas color coded in red and orange displayed in Figure 3), where the blood supply is rich, and if the orthodontic appliance is discontinuous/for a short time, due to tissue’s adaptability, even 1.2 N could not produce significant tissular damages. As for the intrusion and extrusion movements in reduced periodontium, a higher stress (color coded in different shades of yellow) was visible in the entire PDL, with the highest amount of stress present in the cervical third (2.3–2.8 times higher than MHP), but smaller than for the other three movements (Table 3 and Figure 3), and seeming not to determine ischemic risks if they are discontinuous or applied for a short period of time. This approach seems to agree with some of the clinical reports regarding small orthodontic forces of about 1 N (100–120 g) [9,10,11,12,13,14,18,19], but contradict Wu et al. reports [2,3,16] (employing hydrostatic pressure criteria based on the Ogden hyper-elastic model, design for hyper elastic rubber and not suited for the study of PDL). However, if the clinical situation expressly demands that no risks are allowed for the cervical third of reduced PDL (rotation and translations seeming to be the most damaging), the amount of applied force should be reduced to 0.15–0.4 N (0.2–0.4 N for tipping, 0.15–0.2 N for rotation, and 0.15–0.25 N for translation), in agreement with our previous findings [1,6,7] and other reported results [7,8,14,15,16,34] (of 0.1–0.5 N). The Tresca criterion confirmed our relationships and patterns from our previous reports [1], proving to be a viable and more accurate method of analysis than von Mises.

Our quantitative values corroborate to those obtained by Toms et al. [25] (lower premolar mesh, 5205 nodes/1674 elements, VM, intact periodontium, 1 N, extrusion, 8 KPa apical, and 7.75 KPa cervical), Merdji et al. [33] (lower 1st molar mesh, 557,974 elements of 0.25–1 mm, VM, intact periodontium, 10 N, intrusion, 29.48 KPa apical; 3 N, tipping, 8.96 KPa apical; 3 N, translation, and 6.78 KPa apical) and Shaw et al. [24] (upper incisor mesh, 20,582 nodes/11,924 elements, VM, intact periodontium, apical stress, extrusion/intrusion 2 KPa, and tipping 1 KPa), all employing the adequate [20] failure criteria for PDL. Other studies [4,5,22,23,25,26,27] employing inappropriate [20] failure criteria (maximum principal S1 stress, minimum principal S3 stress, and hydrostatic pressure) reported higher amounts of stresses without comparison and/or correlation with MHP.

The differences between the herein quantitative results and other studies’ [2,3,4,5,16,17,24,25,32,33] results may derive from the 3D models (maxillary incisor [24], canine [2,3,16], premolar [2,3,4,5,16,17,32] or 1st molar [33] vs. lower 2nd premolar), complexity of the models (increased vs. reduced [24,25,33] mesh number of nodes and elements), multi-forces vs. a single/two [2,3,4,5,16,17,24,25,32,33] force, and never be applied in a coherent simulation of horizontal periodontal breakdown [1,6,7,8].

A combined clinical in vitro reports [4,5] (employing hydrostatic pressure failure criterion inappropriate for the study of PDL, and with no correlation with the MHP) reported 0.5–1 N intrusion producing external apical and middle third root resorption [with amount of stress of 9.95 TPa (approximately 9,970,000,000 KPa) vs. 2–16 KPa of the physiological MHP], in strong disagreement with the herein results, our previous [1,6,7] observations, and a recent review article [10]. Our herein and previous studies’ [1,6] results (using similar boundary condition and adequate failure criterion-VM [1,6] and Tresca [1]) in intact and reduced periodontium, suggested that intrusion (0.2 N) is the least invasive movement (with amounts of stresses lower that MHP, affecting mostly the cervical and middle third of PDL, and with no risk of external root resorption, while the rotation being the strongest one and with the highest risk, opposing the Minch et al. study [34] (with intrusion reported to be more invasive). Moreover, our simulations showed that the highest risk of external root resorption and further periodontal breakdown, for all five orthodontic movements, appears to be in the cervical third of PDL (in the first mm, with red and orange color-coded areas—Figure 2 and Figure 3), in agreement with a recent review article [10] and our previous reports [1,6,7]. The elastic analyzes the first order by using extremely low amounts of forces (up to 1.2 N) and maintaining the boundary conditions, displayed proportionality between loads and displacements, respectively, between stresses and deformations, allowing further simulations with a higher amount of force.

For a better accuracy of results, our eighty-one 3D models involved in the simulations included the neurovascular bundle reconstructed in the apical third of the PDL (as in Figure 1), an aspect that was not found in the models of PDL from the published research literature. Neither the approach examining the relationships between the circulatory vessels from PDL, maximum hydrostatic pressure, orthodontic movements, and stressed areas were available/found. In an orthodontic movement, all the anatomical parts are involved, playing a more or less important role in absorbing orthodontic forces and the speed of movement; thus, the correlations between these parts should be examined and considered when interpreting the results. In clinical practice, an orthodontic movement is often a combination of several other movements, thus knowing the involved areas of each pure movement and the circulatory vessel’s anatomy is important for predicting the results and potential risks.

When analyzing FEA reports, it must be kept in mind that the accuracy depends on the proper selection of the failure criterion and the anatomical accuracy of the models and their physical properties [1,6,7,20,21]. It must be acknowledged that there are also differences (i.e., not quantitatively significant) between the results of similar FEA simulations (e.g., small changes in the surface of the applied force), nonetheless, the scientific pattern should be kept, and the conclusions should remain similar [1,2,3,6,7,16,25,26,27]. Thus, when analyzing an FEA simulation, the first step is to investigate if the selected failure criterion was designed for the type of material to be analyzed (brittle, liquid/gas or ductile) [1,6,7,20].

There is a wide variety of PDL studies [2,3,4,5,16,17,22,23,24,25,26,27,28,29,30,31,32] with different results, which often contradict the clinical data and hence the mistrust with which the FEA simulations are viewed [1,6,7]. The main weaknesses of these studies [2,3,4,5,16,17,22,23,24,25,26,27,28,29,30,31,32] relate to the lack of understanding of the FEA method and the theory of yielding of materials [1]. A major issue was the use of improper failure criteria [1] (i.e., design for a certain type of material–brittle [22,23,25,26,27,32] or liquid [2,3,4,5,16,17] and used for analyzing a ductile one [1], such as PDL). Another significant shortcoming was considered to be the missing of the relationship [1,6] between the physiological capillary pressure and the quantitative reports [2,3,4,5,16,17,25,26,27] exceeding the MHP. The minor shortcoming was mainly related to the use of ideal anatomy, and of the 3D simplified and incomplete models (e.g., lacking the dental pulp and/or NVB) [2,3,4,5,16,17,22,23,24,25,26,27,28,29,30,31,32]. Nonetheless, if all FEA analyses requirements are met, the results are exact and in agreement with clinical knowledge [1,6,7,20]. Any anatomical alteration or simplification of the studied models would change the accuracy of the results [6,7,8,16]. The herein study employed eighty-one highly anatomically accurate models (0.08–0.116 mm global element size), and more reliable mesh models (40–12,731 times more C3D4 elements; 4.4–1463 times more nodes) than the previous studies [4,5,22,23,24,25,26,27].

The understanding of the yielding theory behind the FEA functioning algorithm is important for selecting the adequate failure criteria to be used. The hydrostatic pressure (widely used in the study of PDL [2,3,4,5,16,17,32]) was mathematically designed for liquids, where the incompressibility appears, and where the shear stress does not exist. However, the PDL, despite having approximately 80% water during its mechanical functioning, suffers variable levels of shear stress that have significant effects over the circulatory vessels and metabolism [1,6]. The yielding theory states that a material, when subjected to a stress, suffers from variable levels of deformation before its destruction through fracture/rupture. A brittle material (e.g., clay and stone) has negligible deformation before its fracture, while a ductile one meets a significant deformation (i.e., elastic and/or plastic) followed by fracture/rupture [1,6,7]. The PDL has a significant ductile resemblance, but shows also a certain brittle flow behavior [1,6,7]. Thus, this type of behavior is characterized by the fact that under a reduced amount of force, an elastic deformation allows it to return and regain its original shape, but if a higher stress is applied, the deformations do not necessarily spring back, but also do not break. That is why elasticity as a physical property is allowed when investigating the PDL behavior. Moreover, almost all ductile materials (owning different degrees of elasticity) under a force of around 1 N display elastic behavior. For brittle materials, the yielding criterions are maximum and minimum principal stresses, while for ductile materials, Tresca and von Mises. Both von Mises and Tresca criterions are similar for ductile isotropic materials, but with Tresca assessing the critical value of maximum shear stress in the structure opposing to the distortional energy of von Mises. Tresca failure criterion states that yielding occurs when the maximum shear stress is equal to the shear stress at yielding in a uniaxial tensile test. Tresca yield surface lies entirely inside the VM surface, thus being more conservative. The PDL inner structure, with variously orientated dense fiber bundles binding the bone of the cementum, encounters during its functioning variable degrees of shear deformation. Tresca criterion, unlike von Mises (describing a smooth behavior), describes a non-smooth behavior in all three principal coordinate planes, fitting the anatomical structure of PDL, and combining the ductile flow with a brittle fracture one, opposing the classical ductile material model (e.g., steel). The divergence (difference) between Tresca and von Mises is around 15% (for Tresca). Tresca cannot be used for isotropic materials or ductile metals (as steel), opposing to von Mises. In most traditional mechanical studies, von Mises was preferred to Tresca because it agreed better with experimental data. Tresca was sometimes used because it was easier to apply and more conservative by predicting a narrower elastic region. The herein quantitative and qualitative stress values are close to previous reports [1,6] employing the VM criterion and to a simulation with the mostly used five criterions, confirming the suitability of Tresca failure criterion for the study of PDL.

An exact simulation of the anatomical biomechanical behavior of periodontium requires the employment of adequate physical properties and conditions. In vivo tissues own anisotropy, non-homogeneity, and non-linear elasticity [4,5,6,7,8,24,26,27,28,29,30,31]. Nonetheless, almost all FEA studies correctly assume isotropy, homogeneity, and linear elasticity, due to the simplicity of the constitutive equations and to the fact that the difference linear vs. non-linear was reported to be of a maximum 10–30% increase in the quantitative values for the non-linear approach [6,7,8,22,23,25]. Nonetheless, a 30% increase in the quantitative stress values would not change the accuracy of the patterns and conclusions. Additionally, from the mechanical point of view, the linear elasticity behavior is expected for all materials subjected to a force of approximately 1 N [6,7,8]. In vivo, there are no pure orthodontic movements, but often an association and combinations of distinct types of movements. Thus, the amounts of stresses that are displayed in the periodontal structures are expected to be lower than herein results, making our approach of importance for clinical practice. Nevertheless, it must be acknowledged that simulations (despite many benefits) are not clinically correct, and a comprehensive knowledge of all the data must be ensured by performing associations and relationships between the known clinical and in vitro data when interpreting the results [6,7,8].

The present research examined eighty-one models of the 2nd lower premolar and surrounding periodontium from nine patients (nine models/patient) employing the same methodology and sample size of nine as in previous studies [1,6,7,8]. Opposingly to our simulations, previous studies using VM criteria [2,3,4,5,16,17,24,25,32,33] used only one tooth model/sample size of one (i.e., anatomically simplified) in a single simulation (maxillary incisor [24], canine [2,3,16], premolar [2,3,4,5,16,17,32], and 1st molar [33]). Despite a sample size limited to one tooth, their reports [2,3,4,5,16,17,24,25,32,33] agreed with herein both qualitatively and quantitatively. Other studies [4,5,22,23,25,26,27] with the same sample size of one reported a quantitative result higher than the herein, principally due to the inadequate applied failure criteria (i.e., S1, S3, or pressure). Nonetheless, the common aspect for all these studies [2,3,4,5,16,17,22,23,24,25,26,27,32,33], regardless of the criterion used, the number of subjects analyzed, and a sample size of one, is that the qualitative and quantitative reports are similar within the studies using the same failure criterion.

Our study (sample size of nine but with eighty-one simulations, nine times more patients, and eighty-one times more models) maintained the same pattern, reporting qualitative (color-coded projections) results with a similar resemblance for all models, and with minimal differences of the quantitative stress values from one model to another, thus valid conclusions could be drawn. Moreover, due to small differences between quantitative and qualitative reports between studies using the same failure criteria (as mentioned above), a higher number of patients (larger sample size) seem not to radically change the results (color-coded stress distribution in tissues and the average amount of stress) or the conclusions.

However, it must be acknowledged that a FEA simulation could not accurately reproduce a clinical situation due to difficulties related to the anatomical micro-architecture of tissues and force applying. Thus, following the above and considering herein analysis limitations (sample size of nine patients, FEA cannot completely simulate clinical situations), we recommend more studies on this subject (but with more advanced modeling of anatomical micro-architecture of tissues) to enhance the knowledge of this issue. To obtain a better understanding of the relationships, associations, and correlations between orthodontic force, MHP, external root resorption, and periodontal breakdown process, further simulations are needed for the reduced periodontium. Thus, an assessment of the stresses displayed in the dental pulp and neurovascular bundle, and on the external root surfaces subjected to the orthodontic forces and correlated with the hydrostatic pressure in intact and reduced periodontium, is needed.

## 5. Conclusions

Based on the herein simulations and taking into account the methodological advantages and limitations (e.g., eighty-one 3D models of nine patients), valid conclusions could be drawn:The 0.6 N of force could be relatively safe in both intact and up to 8 mm reduced periodontium, with almost no risks of circulatory disturbances, for all five pure orthodontic movements.In intact periodontium, 120 g could be safely used and considered to be maximal for all five movements.In reduced periodontium, 1.2 N, despite being completely safe for the apical vessels blood supply in the apical third of PDL, could generate some minor ischemic risks for the perforating vessels in the middle third after 4–5 mm of bone loss for all five movements.In the cervical third, 120 g of force seems to be generating high ischemic risks for reduced periodontium. Nonetheless, in reduced periodontium for rotation, translation, and tipping, the most affected area is represented by the first 1 mm of height of the PDL (which is rich in blood supply); thus, if the orthodontic appliance is discontinuous, due to tissues adaptability, even 1.2 N could not produce significant tissular damages.If the clinical situation expressly demands that no risks are allowed for the cervical third of reduced PDL (rotation and translations seeming to be the most damaging), the amount of applied force should be reduced to 0.15–0.4 N (0.2–0.4 N for tipping, 0.15–0.2 N for rotation, and 0.15–0.25 N for translation).Tresca failure criterion seems more adequate in the FEA study of PDL.

## 6. Practitioner Points

In intact periodontium, 0.6–1.2 N could be safely used and considered to be maximal/optimal for all five pure orthodontic movements. In reduced periodontium, 0.6 N is safe for up to 4 mm of loss. For 4–8 mm of tissue loss, for avoiding any circulatory and resorptive risks, the force should be reduced to 0.2–0.4 N. However, if some risks are acceptable, a 1.2 N discontinuous force could be used for up to 8 mm of periodontal breakdown for all five movements due to the limited areas of high stress displayed in the first mm of height of cervical PDL, with no significant expected damages. It must be acknowledged that these results were reported for the five pure movements. Nevertheless, clinically, 1.2 N of force, due to associations of movements, could produce quantitative stresses in the cervical third of PDL even smaller than the herein and no circulatory and/or resorptive risks. Thus, the clinical approach of a case with reduced periodontium must be assessed individually.

## Figures and Tables

**Figure 1 ijerph-20-01889-f001:**
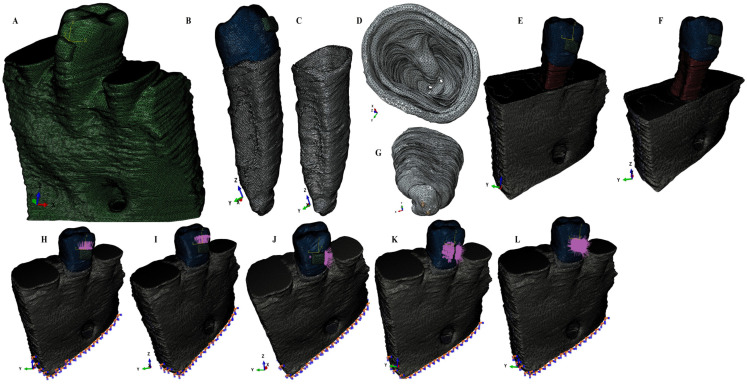
Mesh model of one of the nine patients involved in the study: (**A**)—intact periodontium mesh model, (**B**)—2nd lower right premolar model with intact periodontium and applied bracket, (**C**)—intact PDL: (**D**)—cervical third view of the intact PDL, (**E**)—4 mm bone loss, (**F**)—8 mm bone loss; (**G**)—apical third view of the intact PDL with its neuro-vascular bundle; applied loads vectors: (**H**)—intrusion, (**I**)—extrusion, (**J**)—translation, (**K**)—rotation, and (**L**)—tipping.

**Figure 2 ijerph-20-01889-f002:**
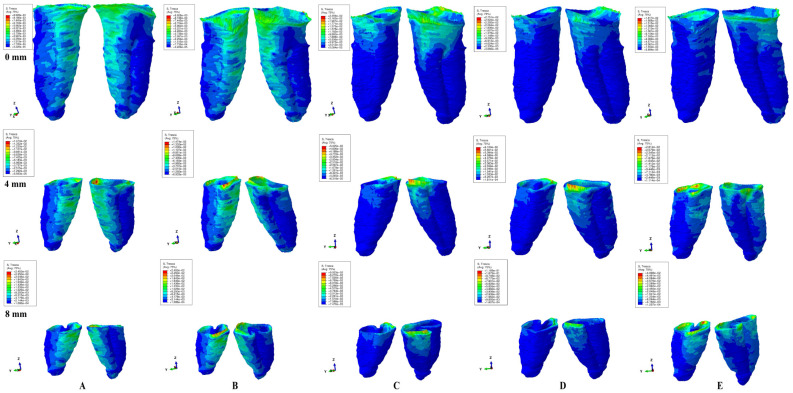
Tresca shear stress display in PDL for 0.6 N (the quantitative values are in MPa) (intact, 4 mm and 8 mm reduced periodontium—vestibular-distal and mesial-lingual view): (**A**)—intrusion, (**B**)—extrusion, (**C**)—translation, (**D**)—rotation, and (**E**)—tipping; the highest stressed areas are color coded in red and orange.

**Figure 3 ijerph-20-01889-f003:**
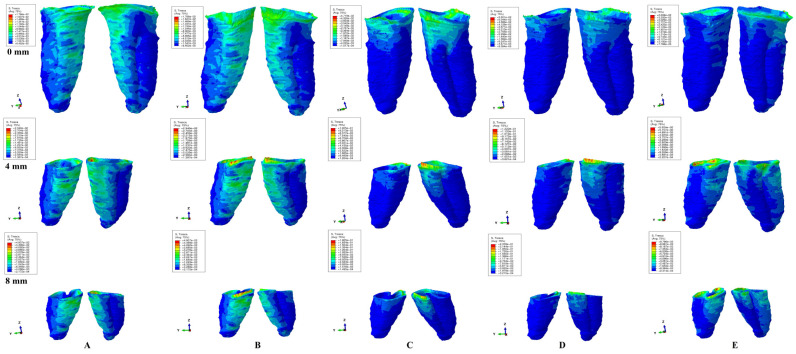
Tresca shear stress display in the PDL for 1.2 N (the quantitative values are in MPa) (intact, 4 mm and 8 mm reduced periodontium—vestibular-distal and mesial-lingual view): (**A**)—intrusion, (**B**)—extrusion, (**C**)—translation, (**D**)—rotation, and (**E**)—tipping the highest stressed areas are color coded in red and orange.

**Table 1 ijerph-20-01889-t001:** Elastic properties of materials in GPa.

Material	Young’s Modulus, E (GPa)	Poisson Ratio, ʋ	Refs.
Enamel	80	0.33	[1,6,7,8]
Dentin/Cementum	18.6	0.31	[1,6,7,8]
Pulp	0.0021	0.45	[1,6,7,8]
PDL	0.0667	0.49	[1,6,7,8]
Cortical bone	14.5	0.323	[1,6,7,8]
Trabecular bone	1.37	0.3	[1,6,7,8]
Bracket (Cr-Co)	218	0.33	[1,6,7,8]

**Table 2 ijerph-20-01889-t002:** The Tresca shear stress average values (KPa) produced by 0.6 N of orthodontic forces.

Resorption (mm)		0	1	2	3	4	5	6	7	8
Intrusion 0.6 N	a	**3.00**	**3.49**	**3.98**	**4.46**	**4.96**	**5.78**	**6.62**	**7.43**	**8.25**
	% a	1.00	1.16	1.33	1.49	1.65	1.93	2.21	2.48	2.75
	m	**3.00**	**5.21**	**5.51**	**6.31**	**7.41**	**7.62**	**7.83**	**8.04**	**8.25**
	% m	1.00	1.74	1.84	2.10	2.47	2.54	2.61	2.68	2.75
	c	**5.22**	**6.68**	**8.15**	**9.61**	**11.07**	**12.91**	**14.75**	**16.58**	**18.43**
	% c	1.00	1.28	1.56	1.84	2.12	2.47	2.83	3.18	3.53
Extrusion 0.6 N	a	**3.00**	**3.80**	**4.59**	**5.39**	**6.18**	**6.70**	**7.22**	**7.73**	**8.25**
	% a	1.00	1.27	1.53	1.80	2.06	2.23	2.41	2.58	2.75
	m	**3.00**	**3.80**	**4.59**	**5.39**	**6.18**	**6.70**	**7.22**	**7.73**	**8.25**
	% m	1.00	1.27	1.53	1.80	2.06	2.23	2.41	2.58	2.75
	c	**6.70**	**8.41**	**10.11**	**11.82**	**13.52**	**15.77**	**18.04**	**20.29**	**22.55**
	% c	1.00	1.25	1.51	1.76	2.02	2.35	2.69	3.03	3.36
Translation 0.6 N	a	**2.01**	**2.57**	**3.13**	**3.69**	**4.25**	**5.06**	**5.88**	**6.69**	**7.51**
	% a	1.00	1.28	1.56	1.84	2.11	2.52	2.93	3.33	3.74
	m	**3.97**	**5.09**	**6.20**	**7.31**	**8.43**	**10.09**	**11.76**	**13.43**	**15.10**
	% m	1.00	1.28	1.56	1.84	2.12	2.54	2.96	3.38	3.80
	c	**17.71**	**23.76**	**29.80**	**35.84**	**41.89**	**48.29**	**54.69**	**61.10**	**67.70**
	% c	1.00	1.34	1.68	2.02	2.37	2.73	3.09	3.45	3.82
Rotation 0.6 N	a	**2.34**	**3.07**	**3.80**	**4.53**	**5.26**	**6.42**	**7.58**	**8.73**	**9.89**
	% a	1.00	1.32	1.63	1.94	2.25	2.75	3.24	3.74	4.24
	m	**4.62**	**6.05**	**7.48**	**8.90**	**10.33**	**12.65**	**14.98**	**17.30**	**19.62**
	% m	1.00	1.31	1.62	1.93	2.23	2.74	3.24	3.74	4.24
	c	**18.35**	**25.23**	**32.11**	**38.98**	**45.86**	**51.47**	**57.07**	**62.68**	**68.28**
	% c	1.00	1.37	1.75	2.12	2.50	2.80	3.11	3.42	3.72
Tipping 0.6 N	a	**1.55**	**2.36**	**3.16**	**3.96**	**4.78**	**5.65**	**6.52**	**7.39**	**8.26**
	% a	1.00	1.52	2.04	2.55	3.08	3.64	4.21	4.77	5.33
	m	**3.06**	**4.07**	**5.09**	**6.10**	**7.11**	**8.42**	**9.73**	**11.03**	**12.34**
	% m	1.00	1.33	1.66	1.99	2.32	2.75	3.18	3.60	4.03
	c	**12.13**	**15.55**	**18.96**	**22.38**	**25.79**	**30.57**	**35.35**	**40.13**	**44.91**
	% c	1.00	1.28	1.56	1.84	2.13	2.52	2.91	3.31	3.70

a—apical stress, m—middle stress, c—stress cervical, % a—nr. of times of stress increase apically, % m—nr. of times of stress increase medially, and % c—nr. of times of stress increase cervically.

**Table 3 ijerph-20-01889-t003:** The Tresca shear stress average values (KPa) produced by 1.2 N of orthodontic force.

Resorption (mm)		0	1	2	3	4	5	6	7	8
Intrusion 1.2 N	a	**5.99**	**6.98**	**7.95**	**8.93**	**9.92**	**11.55**	**13.23**	**14.85**	**16.50**
	% a	1.00	1.17	1.33	1.49	1.66	1.93	2.21	2.48	2.75
	m	**5.99**	**10.41**	**11.01**	**12.62**	**14.81**	**15.24**	**15.66**	**16.08**	**16.50**
	% m	1.00	1.74	1.84	2.11	2.47	2.54	2.61	2.68	2.75
	c	**10.44**	**13.37**	**16.30**	**19.22**	**22.15**	**25.82**	**29.49**	**33.17**	**36.86**
	% c	1.00	1.28	1.56	1.84	2.12	2.47	2.82	3.18	3.53
Extrusion 1.2 N	a	**5.99**	**7.59**	**9.18**	**10.77**	**12.37**	**13.40**	**14.43**	**15.47**	**16.50**
	% a	1.00	1.27	1.53	1.80	2.06	2.23	2.41	2.58	2.75
	m	**5.99**	**7.59**	**9.18**	**10.77**	**12.37**	**13.40**	**14.43**	**15.47**	**16.50**
	% m	1.00	1.27	1.53	1.80	2.06	2.23	2.41	2.58	2.75
	c	**13.41**	**16.81**	**20.22**	**23.63**	**27.04**	**31.54**	**36.07**	**40.59**	**45.10**
	% c	1.00	1.25	1.51	1.76	2.02	2.35	2.69	3.03	3.36
Translation 1.2 N	a	**4.03**	**5.15**	**6.26**	**7.38**	**8.49**	**10.13**	**11.75**	**13.39**	**15.02**
	% a	1.00	1.28	1.55	1.83	2.11	2.51	2.92	3.32	3.73
	m	**7.95**	**10.18**	**12.40**	**14.63**	**16.85**	**20.19**	**23.53**	**26.86**	**30.20**
	% m	1.00	1.28	1.56	1.84	2.12	2.54	2.96	3.38	3.80
	c	**35.43**	**47.52**	**59.60**	**71.69**	**83.77**	**96.58**	**109.39**	**122.19**	**135.40**
	% c	1.00	1.34	1.68	2.02	2.36	2.73	3.09	3.45	3.82
Rotation 1.2 N	a	**4.67**	**6.14**	**7.60**	**9.06**	**10.51**	**12.83**	**15.15**	**17.47**	**19.79**
	% a	1.00	1.31	1.63	1.94	2.25	2.75	3.24	3.74	4.24
	m	**9.25**	**12.10**	**14.95**	**17.81**	**20.67**	**25.31**	**29.95**	**34.60**	**39.24**
	% m	1.00	1.31	1.62	1.93	2.24	2.74	3.24	3.74	4.24
	c	**36.70**	**50.46**	**64.21**	**77.97**	**91.72**	**102.93**	**114.14**	**125.35**	**136.56**
	% c	1.00	1.37	1.75	2.12	2.50	2.80	3.11	3.42	3.72
Tipping 1.2 N	a	**3.10**	**4.72**	**6.32**	**7.92**	**9.56**	**11.30**	**13.04**	**14.78**	**16.53**
	% a	1.00	1.52	2.04	2.55	3.08	3.65	4.21	4.77	5.33
	m	**6.12**	**8.15**	**10.17**	**12.20**	**14.22**	**16.84**	**19.45**	**22.07**	**24.68**
	% m	1.00	1.33	1.66	1.99	2.32	2.75	3.18	3.60	4.03
	c	**24.26**	**31.09**	**37.92**	**44.75**	**51.58**	**61.14**	**70.70**	**80.26**	**89.82**
	% c	1.00	1.28	1.56	1.84	2.13	2.52	2.91	3.31	3.70

a—apical stress, m—middle stress, c—stress cervical, % a—nr. of times of stress increase apically, % m—nr. of times of stress increase medially, and % c—nr. of times of stress increase cervically.

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
