# Peer review of "Assessment of the Maximum Amount of Orthodontic Force for PDL in Intact and Reduced Periodontium (Part I)"

_ijerph, 2023, doi:10.3390/ijerph20031889_

Round 1

Reviewer 1 Report

The aim of the study was to examine 0.6 N and 1.2 N to be the best amount of orthodontic force for PDL at multiple levels of periodontal breakdown, and the relationships with the ischemic risks of further tissue loss and type of movement. The second aim was to examine if Tresca FEA failure criteria is more adequate for the in vitro study of periodontal ligament.

The topic of the study is interesting and relevant.

In the Introduction the Authors underlined the importance of the research. The aim of the study has been precisely defined.

The methodology applied has been discussed in sufficient details. However, please provide more information about the characteristics of the study participants e.g. age, gender etc.

The results are quite clearly presented.

In the Discussion the Authors discussed the obtained results and compared with other findings. However, they also should discussed the limitations of the study.

Minor remarks:

p. 1, line 16 – Please use “Eighty-one 3D models” at the beginning of the sentence instead of “81 3D models”

It should be: “CBCT” instead of “CBTC” p. 3, line 106

Author Response

Department of Cariology, Endodontics and Oral Pathology

Faculty of Dental Medicine

University of Medicine and Pharmacy

Dr. Taylor Tao

Assigned Editor

Special Issue- Advances of Digital Dentistry and Prosthodontics 

                                                                                                                  December 30, 2022

Dear Dr. Taylor Tao,

Thank you very much for your letter dated December 26, 2022, with the comments of the reviewers. We have now carefully considered the comments of the reviewers and amended the paper accordingly. All changes are highlighted in red throughout the manuscript and included also below.

Reply to Reviewer #1:

We agree and we thank the reviewer for his/her time and comments. Appropriate changes in the manuscript have by now been made. Please see below and in the manuscript.

1.Concern of the reviewer:

“The methodology applied has been discussed in sufficient details. However, please provide more information about the characteristics of the study participants e.g. age, gender etc.”

Our response:

  • We thank the reviewer for his/her concern and comments. We do hope that our changes are according to the reviewer‘s remarks.

Revised text: Methodology section pg:4 lines 163-173

“This analysis was performed over nine patients (4 males, mean age 29.81 ± 1.45 years, and informed oral consent) with reduced noninflamed periodontium (i.e., treated chronic periodontitis/ stage II/III grade B periodontitis enrolled in supportive periodontal therapy). Despite the fact that more patients were considered for this research, only nine met the eligibility criteria: reduced noninflamed periodontium, complete mandibular arches (no teeth loss), various levels of bone loss, indication for orthodontic treatment and availability of follow-up throughout treatment. Area of focus was x-ray analyzed (CBCT/cone-beam computed tomography - ProMax 3DS-Planmeca, Finland; voxel size -0.075 mm) and included the lower premolars and first and second molars.”

2.Concern of the reviewer:

“In the Discussion the Authors discussed the obtained results and compared with other findings. However, they also should discussed the limitations of the study.”

Our response:

  • We thank the reviewer for his/her concern and comments. We do hope that our changes are according to the reviewer‘s remarks.

Revised text: Discussion section, pg:12 lines 437-454

“An exact simulation of the anatomical biomechanical behavior of periodontium requires the employment of adequate physical properties and conditions. In vivo tissues own anisotropy, non-homogeneity, and non-linear elasticity [4-8, 24, 26-31]. Nonetheless, almost all FEA studies correctly assume isotropy, homogeneity and linear elasticity due to the simplicity of the constitutive equations and to the fact that the difference linear vs. non-linear has been reported to be of maximum 10-30% increase of the quantitative values for the non-linear approach [6-8, 22, 23, 25]. Nonetheless, a 30% increase of the quantitative stress values would not change the accuracy of the patterns and conclusions. Additionally, from the mechanically point of view, the linear elasticity behavior is expected for all materials subjected to a force of approximately 1 N [6-8]. In vivo there are no pure orthodontic movements but often an association and combinations of distinct types of movements. Thus, the amounts of stresses that are displayed in the periodontal structures are expected to be lower than here results, making our approach of importance for clinical practice. Nevertheless, it must be acknowledged that simulations (despite many benefits) are not clinically correct, and a comprehensive knowledge of all the data must be ensured by performing associations and relationships between the know clinical and in vitro data when interpreting the results [6-8].”

                   Practitioner point, pg.13, lines 483-494

“In intact periodontium 0.6-1.2 N could be safely used and considered to be maximal/optimal for all five pure orthodontic movements. In reduced periodontium 0.6 N is safe up to 4 mm of loss. For 4-8 mm of tissue loss for avoiding any circulatory and resorptive risks, the force should be reduced to 0.2-0.4 N. However, if some risks are acceptable 1.2 N discontinuous force could be used up to 8 mm of periodontal breakdown for all five movements due to the limited areas of high stress displayed in the first mm of height of cervical PDL, with no significant expected damages. It must be acknowledged that these results were reported for the five pure movements. Nevertheless, clinically 1.2 N of force, due to associations of movements, could produce quantitative stresses in the cervical third of PDL even smaller than the herein and no circulatory and/or resorptive risks. Thus, the clinical approach of a case with reduced periodontium must be assessed individually.”

3.Concern of the reviewer:

“p. 1, line 16 – Please use “Eighty-one 3D models” at the beginning of the sentence instead of “81 3D models.”

Our response:

  • We thank the reviewer for his/her concern and comments. We do hope that our changes are according to the reviewer‘s remarks.

Revised text: Abstract section, pg:1, lines 16-19

“Eighty-one 3D models (from nine patients; nine models/patient) with the 2nd lower premolar and different degrees of bone loss (0-8 mm) where subjected to continuous intrusion, extrusion, rotation, translation, and tipping movements”

4.Concern of the reviewer:

“It should be: “CBCT” instead of “CBTC” p. 3, line 106.”

Our response:

  • We thank the reviewer for his/her concern and comments. We do hope that our changes are according to the reviewer‘s remarks.

Revised text: Abstract section, pg:3, lines 113-115

“This method is based on CBCT (cone-beam computed tomography) investigations of patients with various levels of periodontal breakdown, and widely used in clinical practice.”

Reviewer 2 Report

The article presented to me for evaluation concenrs the determination of the optimal forces exerted on the teeth during orthodontic treatment, which can be safely used in conditions of healthy periodontium and periodontal breakdown from 1 to 8 mm. In the study, the authors used the Finite Element Analysis to determine the maximum orthodontic forces that can be exerted during treatment.

The introduction clearly describes the current state of knowledge in the field of the analyzed problem. The purpose of the research is clearly formulated. The research methodology is sufficiently described. In this part of the article, the authors drew attention to the limitations of the Finite Element Analysis method and the need to determine the boundary conditions.

The results are clearly summarized in tables and illustrated in figures. The extensive discussion justifies the conclusions formulated in the article.

The subheading "Practitioner Points" in which the authors presented conclusions interesting from the point of view of a clinician dealing with orthodontic treatment, is a valuable part of the article.

English should be improved throughout the article, in some places the sentences are incomprehensible.

To sum up, the article can be published after linguistic corrections has been made. It will certainly be of interest to orthodontists and other specialists dealing with treatment of the stomatognathic system and engineers dealing with FEA in biological systems.

Author Response

Department of Cariology, Endodontics and Oral Pathology

Faculty of Dental Medicine

University of Medicine and Pharmacy

Dr. Taylor Tao

Assigned Editor

Special Issue- Advances of Digital Dentistry and Prosthodontics 

                                                                                                                  December 30, 2022

Dear Dr. Taylor Tao,

Thank you very much for your letter dated December 26, 2022, with the comments of the reviewers. We have now carefully considered the comments of the reviewers and amended the paper accordingly. All changes are highlighted in red throughout the manuscript and included also below.

Reply to Reviewer #2:

We agree and we thank the reviewer for his/her time and comments. Appropriate changes in the manuscript have by now been made. Please see below and in the manuscript.

1.Concern of the reviewer:

“English should be improved throughout the article, in some places the sentences are incomprehensible.

To sum up, the article can be published after linguistic corrections has been made. It will certainly be of interest to orthodontists and other specialists dealing with treatment of the stomatognathic system and engineers dealing with FEA in biological systems.”

Our response:

  • We thank the reviewer for his/her concern and comments. We do hope that our changes are according to the reviewer‘s remarks.

Revised text: Entire manuscript

Reviewer 3 Report

Dear authors,

I evaluated the article titled “Assessment of the Optimal Amount of Orthodontic Force for PDL in Intact and Reduced Periodontium (Part I)”; Goal: “examine 0.6 N and 1.2 N to be the best amount of orthodontic force for PDL at multiple levels of periodontal breakdown, and the relationships with the ischemic risks of further tissue loss and type of movement”.

The theme is very interesting but, particularly, I do not like of "salami" studies, only in case of very long studies (it was not the case). I suggest the authors merge Part II here. Moreover, there was a low number of patients included, which impaired any conclusion.

- There is necessity of English language review;

Abstract: it is incomplete.

INTRO

- It was justified the forces chosen and used in the study;

- I considered the intro long, but it is well-written.

M&M: I appreciate the M&M, but I have some comments (below)

- I had a question: Was it chosen the forces (0.6 or 1.2N) or the authors still want to find them? (lines 147-149: “It is trying to find the best orthodontic force applicable for the apical and cervical third of PDL without major ischemic/necrotic risks.”); In this case, I suggest to applied forces like 0.5N and 1.4N (e.g.) - but for periodontal patients, it can be a risk

(high forces)

- why was included only nine patients? Where is the sample size calculation?

- Eligibility criteria is very poorly presented

RESULTS and DISCUSSSION: I considered well-presented.

CONCLUSION: it is impossible to do a strong conclusion with the low number of samples.

Author Response

Department of Cariology, Endodontics and Oral Pathology

Faculty of Dental Medicine

University of Medicine and Pharmacy

Dr. Taylor Tao

Assigned Editor

Special Issue- Advances of Digital Dentistry and Prosthodontics 

                                                                                                                  December 30, 2022

Dear Dr. Taylor Tao,

Thank you very much for your letter dated December 26, 2022, with the comments of the reviewers. We have now carefully considered the comments of the reviewers and amended the paper accordingly. All changes are highlighted in red throughout the manuscript and included also below.

Reply to Reviewer #3:

We agree and we thank the reviewer for his/her time and comments. Appropriate changes in the manuscript have by now been made. Please see below and in the manuscript.

1.Concern of the reviewer:

“The theme is very interesting but, particularly, I do not like of "salami" studies, only in case of very long studies (it was not the case). I suggest the authors merge Part II here. Moreover, there was a low number of patients included, which impaired any conclusion.

- There is necessity of English language review;

Abstract: it is incomplete.”

Our response:

  • We thank the reviewer for his/her concern and comments. We do hope that our changes are according to the reviewer‘s remarks.

The second part of the study is another totally different manuscript (as complex as the present one) that examines the dental pulp and its neuro-vascular bundle. The “Part I” and “Part II” was seen as a help for the reader (to communicate that both type of tissues are studied).

Revised text: Entire manuscript- English language review

                       Abstract section, pg:1, lines 14-29

“This study examines 0.6N and 1.2N as the maximum orthodontic force for periodontal ligament (PDL) at multiple levels of periodontal breakdown, and the relationships with the ischemic, necrotic and resorptive risks. Additionally, evaluates if Tresca failure criteria is more adequate for the PDL study. Eighty-one 3D models (from nine patients; nine models/patient) with the 2nd lower premolar and different degrees of bone loss (0-8 mm) where subjected to continuous intrusion, extrusion, rotation, translation, and tipping movements. Tresca shear stress was assessed individually for each movement and bone loss level. Rotation and translation produced the highest stresses, while intrusion and extrusion determined the lowest. Apical and middle third stresses were lower than the cervical stress. In intact periodontium, shear stress was lower than the maximum hydrostatic pressure (MHP) for both forces. In reduced periodontium (1-8 mm tissue loss), the apical amount of shear stress was lower than MHP for both forces, while cervically for rotation, translation and tipping movements exceeded the 16KPa. 1.2N could be used both in intact periodontium and for the apical and middle third of PDL up to 8 mm of bone loss without any risks. However, for avoiding any resorptive risks, in the cervical third of PDL, the rotation, translation and tipping movements require less than 0.2-0.4N of force after 4mm of loss. Tresca seems to be more adequate for the study of PDL than other criteria.”

2.Concern of the reviewer:

“M&M: I appreciate the M&M, but I have some comments (below)

- I had a question: Was it chosen the forces (0.6 or 1.2N) or the authors still want to find them? (lines 147-149: “It is trying to find the best orthodontic force applicable for the apical and cervical third of PDL without major ischemic/necrotic risks.”); In this case, I suggest to applied forces like 0.5N and 1.4N (e.g.) - but for periodontal patients, it can be a risk

(high forces)

- why was included only nine patients? Where is the sample size calculation?

- Eligibility criteria is very poorly presented.”

Our response:

  • We thank the reviewer for his/her concern and comments. We do hope that our changes are according to the reviewer‘s remarks.

Revised text:  Introduction section, pg. 2, lines 81-88

“A report of our group found the rotation movement to be the most invasive along with translation, and that 0.5 N is safe to be used for the cervical third of PDL in intact periodontium, while the amount of force should be reduced to 0.1–0.2 N for rotation, 0.15–0.3 N for translation and 0.2–0.4 N for tipping in 4–8 mm of periodontal breakdown [1]. However, in the apical third of PDL the same study [1] reported 0.5 N to be safely applied up to 8 mm of periodontal loss.  Nonetheless, the questions about the maximum orthodontic force to be safely used in apical third of PDL at various levels of bone height without any ischemic and/or necrotic risks remains.”

                                                 pg.3-4, lines 151-155

“This study examines if 0.6 N and 1.2 N could be the maximum amount of orthodontic force for PDL at multiple levels of periodontal breakdown, and the relationships with the ischemic and resorptive risks, further tissue loss and the type of movement. As a second aim, examines if Tresca FEA failure criteria is more adequate for the in vitro study of periodontal ligament.”

                       Material and methods, pg:4, lines 157-173

“The herein analysis is part of a bigger project (clinical protocol 158/02.04.2018) developed progressively in phases with the clear goal to study the lower premolar and the surrounding periodontium at different bone heights under orthodontic forces [1, 6-8]. This study is focused on finding the maximum amount of orthodontic force to be applied for the apical and cervical third of PDL without any major ischemic/necrotic risks.

This analysis was performed over nine patients (4 males, mean age 29.81 ± 1.45 years, and informed oral consent) with reduced noninflamed periodontium (i.e., treated chronic periodontitis/ stage II/III grade B periodontitis enrolled in supportive periodontal therapy). Despite the fact that more patients were considered for this research, only nine met the eligibility criteria: reduced noninflamed periodontium, complete mandibular arches (no teeth loss), various levels of bone loss, indication for orthodontic treatment and availability of follow-up throughout treatment. Area of focus was x-ray analyzed (CBCT/cone-beam computed tomography - ProMax 3DS-Planmeca, Finland; voxel size -0.075 mm) and included the lower premolars and first and second molars.”

3.Concern of the reviewer:

“CONCLUSION: it is impossible to do a strong conclusion with the low number of samples.”

Our response:

  • We thank the reviewer for his/her concern and comments. We do hope that our changes are according to the reviewer‘s remarks.

Revised text: Conclusions section, pg:12 lines 462-468

“Based on the herein simulations and taking into account the methodological advantages and limitations, some conclusions can still be drawn:

  1. 0.6 N of force could be relatively safe in both intact and up to 8 mm reduced periodontium, with almost no risks of circulatory disturbances, for all five pure orthodontic movements.
  2. In intact periodontium 120 g could be safely used and considered to be maximal for all five movements.”

Round 2

Reviewer 3 Report

Dear authors,

I evaluated the article titled “Assessment of the Maximum Amount of Orthodontic Force for PDL in Intact and Reduced Periodontium (Part I)”; Goal: “examine 0.6 N and 1.2 N to be the best amount of orthodontic force for PDL at multiple levels of periodontal breakdown, and the relationships with the ischemic risks of further tissue loss and type of movement”.

The theme is very interesting and I understood about Part I and Part II.

Abstract: it could be better presented (results)

M&M: I appreciate the M&M, but I have yet some concerns (below)

- “study is focused on finding the maximum amount of orthodontic force”

It was chosen 2 forces. How are the authors intending to find the maximum force if they chosen just 2 (0.6 or 1.2N)?

- 9 patients were included. It is a low sample to find a conclusion. Where is the sample size calculation to justify it?

Moreover, lines 158-159: “over nine patients (4 males, mean age 29.81 ± 1.45 years, and informed oral consent)”. Were 9 or 4 patients included?

- Eligibility criteria is poorly presented

CONCLUSION: it is impossible yet to conclude with the low number of sample

Author Response

Department of Cariology, Endodontics and Oral Pathology

Faculty of Dental Medicine

University of Medicine and Pharmacy

Dr. Taylor Tao

Assigned Editor

Special Issue- Advances of Digital Dentistry and Prosthodontics 

                                                                                                                  January 8, 2023

Dear Dr. Taylor Tao,

Thank you very much for your letter dated January 6, 2023, with the comments of the reviewers. We have now carefully considered the comments of the reviewers and amended the paper accordingly. All changes are highlighted in red throughout the manuscript and included also below.

Reply to Reviewer #3:

We agree and we thank the reviewer for his/her time and comments. Appropriate changes in the manuscript have by now been made. Please see below and in the manuscript.

1.Concern of the reviewer:

“Dear authors,

I evaluated the article titled “Assessment of the Maximum Amount of Orthodontic Force for PDL in Intact and Reduced Periodontium (Part I)”; Goal: “examine 0.6 N and 1.2 N to be the best amount of orthodontic force for PDL at multiple levels of periodontal breakdown, and the relationships with the ischemic risks of further tissue loss and type of movement”.

The theme is very interesting and I understood about Part I and Part II. 

Abstract: it could be better presented (results).”

Our response:

  • We thank the reviewer for his/her concern and comments. We do hope that our changes are according to the reviewer‘s remarks.

   Revised text: Abstract section, pg:1, lines 14-31

“This study examines 0.6N and 1.2N as the maximum orthodontic force for periodontal ligament (PDL) at multiple levels of periodontal breakdown, and the relationships with the ischemic, necrotic and resorptive risks. Additionally, evaluates if Tresca failure criteria is more adequate for the PDL study. Eighty-one 3D models (from nine patients; nine models/patient) with the 2nd lower premolar and different degrees of bone loss (0-8 mm) where subjected to intrusion, extrusion, rotation, translation, and tipping movements. Tresca shear stress was assessed individually for each movement and bone loss level. Rotation and translation produced the highest PDL stresses, while intrusion and extrusion determined the lowest. Apical and middle third PDL stresses were lower than the cervical stress. In intact periodontium, the amount of shear stress produced by the two investigated forces was lower than the 16KPa of the maximum physiological hydrostatic pressure (MHP). In reduced periodontium (1-8 mm tissue loss), the apical amount of PDL shear stress was lower than MHP for both applied forces, while cervically for rotation, translation and tipping movements exceeded the 16KPa. 1.2N could be used in intact periodontium (i.e., without risks) and for the reduced periodontium only in the apical and middle third of PDL up to 8 mm of bone loss. However, for avoiding any resorptive risks, in the cervical third of PDL, the rotation, translation and tipping movements require less than 0.2-0.4N of force after 4 mm of loss. Tresca seems to be more adequate for the study of PDL than other criteria.”

2.Concern of the reviewer:

M&M: I appreciate the M&M, but I have yet some concerns (below)

- “study is focused on finding the maximum amount of orthodontic force”

It was chosen 2 forces. How are the authors intending to find the maximum force if they chosen just 2 (0.6 or 1.2N)?

Our response:

  • We thank the reviewer for his/her concern and comments. We do hope that our changes are according to the reviewer‘s remarks.

The two selected amount of force of 0.6 N and 1.2 N are usually used in the clinical orthodontic therapy and close to the limits of applied forces. In a previous study (reported in the introduction of this manuscript, pg.2, lines 81-88) a simulation with 0.5 N, showed amount of stress lower than the physiological maximum hydrostatic pressure (MHP). Thus, a higher force of 0.6 N was selected to be studied. Furthermore, FEA being a mathematical algorithm-based method (with Tresca failure criteria specially design for), allows a mathematical anticipation/calculation and prediction of PDL behaviour subjected of a higher force of about 1.2 N (and also the forces included in the range mentioned above). These two forces being commonly used by the practitioner we considered to be of interest. If the amount of stress produced by one or both of them exceeds the MHP, implies that no higher force should be used, being the maximum amount to be tolerated by the tissue.     

Revised text:  Material and Methods section, pg. 5, lines 207-214

“If the displayed amount of quantitative stress under one or both investigated orthodontic forces reach or exceed the 16 KPa of the reported maximum physiological hydrostatic pressure, induce a high risk of ischemia, necrosis, and resorption, implying that no higher force should be used, thus being the maximum amount to be tolerated by the tissue. Based on this risk assessment, the simulations with Tresca failure criterion were redone for all models by reducing the amount of orthodontic force applied to the bracket to a level of 0.1–0.4 N, and then the average quantitative results were corelated once more with MHP.”

                         Discussion section, pg. 9, lines 318-327

“The two selected amounts of force of 0.6 N and 1.2 N are usually used in the clinical orthodontic therapy and close to the limits of applied forces. In a previous study [1] a simulation with 0.5 N, showed amount of stress lower than the physiological maximum hydrostatic pressure (MHP). Thus, a higher force of 0.6 N was selected to be studied. Furthermore, FEA being a mathematical algorithm-based method, it allows a mathematical anticipation/calculation and prediction of PDL behavior subjected to a higher force, thus allowing the selection of 1.2 N. These two forces being commonly used by the practitioner in daily practice were considered to be of high interest. If the amount of stress produced by one or both of them exceeds the MHP, implies that no higher force should be used, being the maximum amount to be tolerated by the tissue.”

3.Concern of the reviewer:

- 9 patients were included. It is a low sample to find a conclusion. Where is the sample size calculation to justify it?

Moreover, lines 158-159: “over nine patients (4 males, mean age 29.81 ± 1.45 years, and informed oral consent)”. Were 9 or 4 patients included?

- Eligibility criteria is poorly presented.”

CONCLUSION: it is impossible yet to conclude with the low number of sample.”

Our response:

  • We thank the reviewer for his/her concern and comments. We do hope that our changes are according to the reviewer‘s remarks.

Our present study being a part of a larger research, used the same unchanged methodology (as our previous published analysis- M&M, pg.4, lines157-159), respectively 81 3D models of nine patients, manually reconstructed (implying high accuracy but extremely time consuming) in order to enhance the accuracy of the results. The stress distribution (color-coded projections) in tissues maintained similar resemblance for all models, while the amounts of stress (quantitative) showed small differences from one model to another. Based on these aspects, valid conclusions could be  draw, however we also recommended more studies regarding this aspect for validating or changing our conclusions.

In both the introduction (pg.3, lines 120-144) and discussion sections (pg.10, lines 338-372; pg. 11, lines 394-409) we found that the previous studies related to our subject investigated usually one or two models, from one patient, vs. our study with much more models and patients. Thus, despite a reduced number of patients but with high number of models and simulations (i.e., eighty-one), and acknowledging the limits of our simulations, we disagree regarding the reviewer’s statement (regarding the impossibility to draw conclusion) and based on the approach of our analysis we respectfully consider that valid conclusions could be drawn.

The eligibility criteria included both inclusion and exclusion criterions, that were further detailed.       

Revised text:  Material and Methods section, pg. 4, lines 165-181

“This analysis was performed over eighty-one 2nd lower premolar 3D models from nine patients (4 males and 5 females, mean age 29.81 ± 1.45 years, and informed oral consent) with reduced noninflamed periodontium (i.e., treated chronic periodontitis/ stage II/III grade B periodontitis enrolled in supportive periodontal therapy). Even though more patients were considered for this research, only nine met the inclusion criteria: intact and reduced noninflamed periodontium, proper oral hygiene, complete mandibular arches (no teeth loss for the studied arch), intact second lower premolar (no decays, fillings, endodontic or prosthetic treatment) with adequate anatomical topography (no malposition), various levels of bone loss, indication for orthodontic treatment and availability of follow-up throughout treatment. The exclusion criteria included: intact and reduced inflamed periodontium, poor oral hygiene, tooth loss (lower first molar and premolars), second lower premolar with decays, fillings, prosthetic and endodontic treatment, malposition, inconsistency in the follow-up will. Taking into account that most of the patients who present themselves for orthodontic treatment do not meet the inclusion criteria (due to associated dental troubles) and having in mind that previous studies related to our subject investigated one or two models, from one patient, we considered that by using 81 3D models from 9 patients, valid results and conclusions could be drawn from our study.”

                         Results section, pg. 6, lines 233-236

“The qualitative stress distribution (i.e., color-coded projections) in tissues maintained a similar resemblance for all models, while the amounts of stress (i.e., quantitative values) showed small differences from one model to another.”

                         Discussions section, pg. 12, lines 487-492

“The qualitative (color-coded projections) results in analyzed tissues kept a similar resemblance for all models, and with minimal differences of the quantitative stress values from one model to another, thus valid conclusions could be draw. However, due to limitations (eighty-one models of nine patients, FEA cannot completely simulate clinical situations) we also recommend more studies of this subject to enhance the knowledge of this issue.”

                     Conclusions section, pg:12 lines 500-502

“Based on the herein simulations and taking into account the methodological advantages and limitations (e.g., eighty-one 3D models of nine patients), valid conclusions could be drawn:”
